# OSCILLATION NEURAL ORDINARY DIFFERENTIAL EQUATIONS

## ABSTRACT

Neural ordinary differential equations (NODEs) have received a lot of attention in recent years due to their memory efficiency. Different from traditional deep learning, it defines a continuous deep learning architecture based on the theory of ordinary differential equations (ODEs), which also improves the interpretability of deep learning. However, it has several obvious limitations, such as a NODE is not a universal approximator, it requires a large number of function evaluations (NFEs), and it has a slow convergence rate. We address these drawbacks by modeling and adding an oscillator to the framework of the NODEs. The oscillator enables the trajectories of our model to cross each other. We prove that our model is a universal approximator, even in the original input space. Due to the presence of oscillators, the flows learned by the model will be simpler, thus our model needs fewer NFEs and has a faster convergence speed. We apply our model to various tasks including classification and time series extrapolation, then compare several metrics including accuracy, NFEs, and convergence speed. The experiments show that our model can achieve better results compared to the existing baselines.

## 1 INTRODUCTION

Neural Ordinary Differential Equations (NODEs) (Chen et al., 2018) are the latest continuous deep learning architectures that were first developed in the context of continuous recurrent networks (Cohen & Grossberg, 1983). This continuous deep learning architecture provides a new perspective that theoretically bridges the gap between deep learning and dynamic systems. This deep learning architecture can be efficiently trained with backpropagation and has shown great promise on several tasks including modeling continuous time data, classification, and building normalizing flows.

The core idea of a NODE is to use a neural network to parameterize the vector field (Chen et al., 2018; Kidger, 2022). Typically, a simple neural network is enough to represent the vector field, which will be optimized during the training process. Based on the well-learned vector field, trajectories will be obtained as the estimate functions. However, this architecture has several limitations. First, NODEs cannot learn any crossover-mapping functions (Dupont et al., 2019), which results in them not being universal approximators. Second, to optimize the vector field, it will need many function evaluations during both forward evaluation and backpropagation processes of training. Third, the time and convergence rate of the training process is relatively slow.

The first limitation is caused by the continuity of vector-field-based trajectories because the trajectories in NODEs cannot cross each other at the same time (Massaroli et al., 2020; Norcliffe et al., 2020). This property causes NODEs to be powerless against some special topologies, such as the cases of concentric circles and intersecting lines mentioned by Dupont et al. (2019). We conjecture the reason for the second limitation is caused by the straightforward optimization approach of the vector field. There is no guarantee that learning the vector field is a better choice than learning the estimated functions directly. Sometimes it will need many function evaluations to optimize the vector field, so the difficulties go beyond learning the estimated functions themselves. The third limitation is caused by the trade-off between accuracy and speed for the ordinary differential equation solver (ODE solver). NODEs perform forward evaluation and backpropagation calculations via ODE solvers, which can be treated as black boxes. If we need to ensure the accuracy of an ODE solver, then we must sacrifice the speed.

To address the first problem, we intend to add some discrete elements during the process of learning the trajectories. A discrete element can be realized by a "jump" in the trajectory. This jump allows the trajectories of the NODEs to cross at the same time. It solves the above-mentioned cases. We also give proof in Section 4.1 that our approach is a universal approximator. To solve the second and third problems, we propose to join a function $g$ that directly optimizes the trajectories while optimizing the vector field at the same time. It shifts part of the burden from optimizing the vector field to optimizing the original estimated function. The addition of $g$ will make the final optimized vector fields less complicated, thus a "simple" vector field can be utilized to estimate the function well. This will result in fewer NFEs, less training time, and faster convergence.

Based on the above two points, we design an oscillator to enhance neural ordinary differential equations, namely Oscillation Neural Ordinary Differential Equations (ONODE). The architecture is shown in Figure 1. This oscillator is designed to achieve both "jumping" of trajectories and optimizing the estimation function while simultaneously optimizing the vector field. This design solves the three problems mentioned above to some extent. Our proposed method is not only a universal approximator, but it also reduces the NFEs as well as improves the training convergence speed.

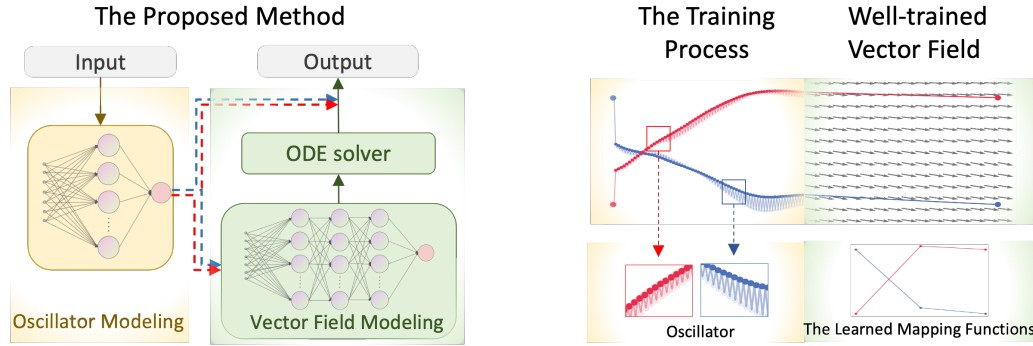

Figure 1: The diagram on the left shows the structure of our proposed model, and the right shows how the oscillator helps our model learn the cross-trajectory by example.

Specifically, the vector field is parameterized by a simple neural network, the same as the NODEs proposed by Chen et al. (2018). The vector field can be optimized by ODE solvers which can be used such as Runge (1895), Kutta (1901), and Hairer et al. (1993). The oscillator we designed is parameterized by a shallow neural network structure and it has only one hidden layer. The oscillator can be located before vector field modeling, or after the ODE solver.

The oscillator modeling structure may vary slightly in different tasks. For example, in the usual classification tasks and extrapolating time series tasks, these structures will have the perception of one hidden layer. In the image classification task, the structure will have two convolutional layers with an activation function. It is worth noting that since our oscillator is parameterized by a shallow neural network that does not take much space, we can still consider the model to be memory efficient. We will illustrate this in Section 5. Besides, our oscillator simplifies the learning complexity of the vector field, which leads to a reduction in NFEs and convergence speed. A simple example is shown in Figure 1 right, the final trained vector field will be a very simple one.

## 2 RELATED WORK

The basic ideas of neural ordinary differential equations were originally considered in Rico-Martinez et al. (1992), Rico-Martinez & Kevrekidis (1993), and Rico-Martinez et al. (1994). For example, Rico-Martinez et al. (1992) proposed a vector field that can be trained using a Multilayer Perceptron (MLP). Rico-Martinez & Kevrekidis (1993) used an implicit integrator and recurrent networks for continuous-time modeling of nonlinear systems. Chen et al. (2018) specified the architecture of NODEs and applied it to several tasks and achieved good results, including image classification, continuous normalizing flows, and latent ODEs, leading to an explosion of interest in NODEs.

NODEs are a new class of model that transform data continuously through infinite depth architectures (Norcliffe et al., 2020). In particular, NODEs can be seen as a continual version of Residual Networks (He et al., 2016) by taking the discretization step to infinitely small. This continuous infinite depth architecture will bring several benefits, one of the most important is memory efficiency. According to Chen et al. (2018), the scalar-valued loss with respect to all inputs of any ODE solver can be computed directly without backpropagating through the operations of the solver. The intermediate quantities of the forward pass will not need to be stored. It causes the NODEs so they can be trained with a constant memory cost.

However, this continuous infinite depth architecture will bring several drawbacks. Dupont et al. (2019) showed that NODEs learn representations that preserve the topology of the input space and prove that the existence of functions in NODEs cannot be represented. To address these limitations, Dupont et al. (2019) introduced Augmented NODEs (ANODEs) which add extra dimensions for the vector field learning. This approach increases the degrees of freedom of the trajectory by elevating the dimensionality, thus circumventing the problem that trajectories cannot be crossed. However, it does not inherently overcome the problem that trajectories cannot be traversed and increases the original parameter space. Other limitations of NODEs include the NFEs which can often become prohibitively large (Dupont et al., 2019) and the speed is slow as long as it is needed to obtain good precision (Chen et al., 2018). These limitations are usually caused by the need for the NODE to learn a complex vector field.

Due to the generality of the NODEs' framework, there is already a lot of related work based on NODEs. Some work focuses on modeling for the time series data. For example, Norcliffe et al. (2020) focused on the learning dynamics system and proposed the Second Order Neural Ordinary Differential Equations (SONODEs). It can be seen as a system of coupled first-order NODEs. Jia & Benson (2019) introduced Neural Jump Stochastic Differential Equations that provided a data-driven approach to learn continuous and discrete dynamic behavior. Kidger et al. (2020) proposed Neural Controlled Differential Equations to model the irregular time series. Morrill et al. (2021) used Neural Rough Differential Equations for modeling long time series. Other works focused on physical modeling. For example, Greydanus et al. (2019) introduced Hamiltonian Neural Networks for learning certain physical laws that follow conservation laws. Cranmer et al. (2020) proposed Lagrangian neural networks to learn certain physical laws without requiring canonical coordinates.

## 3 PRELIMINARY

NODEs are a family of deep neural network models that can be interpreted as a continuous version of Residual Networks (He et al., 2016). Recall the formulation of a residual network:

$$h_{t+1} - h_t = f(h_t, \theta_f), \tag{1}$$

where the $f$ is the residual block and the $\theta_f$ represents the parameters of $f$. The left side of Equation 1 can be seen as the denominator is 1, so it can be represented by $\frac{h_{t+1} - h_t}{1} = f(h_t, \theta_f)$. When the number of layers becomes infinitely large, and the step becomes infinitely small, Equation 1 will become an ordinary differential equation format, as shown in Equation 2.

$$\lim_{dt \to 0} \frac{h_{t+dt} - h_t}{dt} = \frac{dh(t)}{dt} = f(h(t), t, \theta_f). \tag{2}$$

Thus, the NODE will have the same format as an ODE:

$$h'(t) = f(h(t), t, \theta_f), \quad h(0) = \mathbf{x}_0, \tag{3}$$

where $\mathbf{x}_0$ is the input data. Typically, $f$ will be some standard simple neural architecture, such as a MLP. The $\theta_f$ represents trainable parameters in $f$.

To obtain any final state of $h(t)$ when $t = T$, all that is needed is to solve an ordinary differential equation with initial values, which is called an initial value problem (IVP):

$$h(T) = h(0) + \int_0^T f(h(t), t, \theta_f) dt. \tag{4}$$

Thus, a NODE can transform from $h(0)$ to $h(T)$ through the solutions to the initial value problem (IVP) of the ODE. This framework indirectly realizes a functional relationship $x \to F(x)$ like a general neural network.

By the properties of ODEs, NODEs are always invertible; we can reverse the limits of integration, or alternatively, integrate $-f$. The *Adjoint Sensitivity Method* (Pontryagin et al., 1961) based on reverse-time integration of an expanded ODE, allows for finding gradients of the initial value problem solutions $h(T)$ with respect to parameters $\theta_f$ and the initial values $h(0)$. This allows the training NODE to use gradient descent, which allows them to combine with other neural network blocks.

## 4 THE PROPOSED METHODS

### 4.1 OSCILLATION NODE: A UNIVERSAL APPROXIMATOR

As we mentioned, since the trajectories in NODEs cannot cross each other at the same time, a NODE on its own does not have universal approximation capability. An example from Dupont et al. (2019) considers a continuous, differentiable, invertible function $f(x) = -x$ on $\mathcal{X} = \mathbb{R}$. There is no ODE defined on $\mathbb{R}$ that would result in $x_T = \phi_T(x_0) = -x_0$. In ODEs, paths $(x_t, t)$ between the initial value $(x_0, 0)$ and final value $(x_T, T)$ have to be continuous and cannot intersect in $\mathcal{X}$ for two different initial values, and the paths corresponding to $x \to -x$ and $0 \to 0$ would need to intersect.

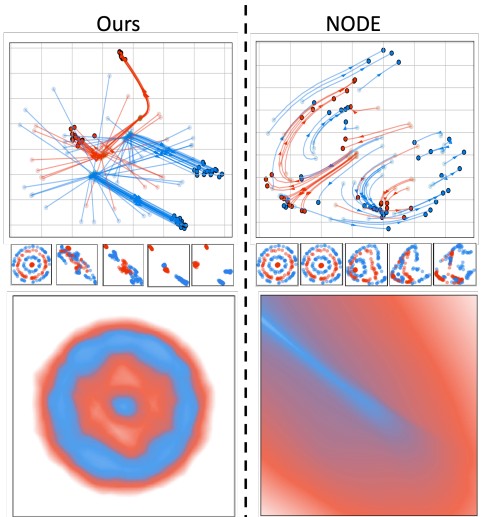

Figure 2: The top two plots show the flow trajectories. The middle layer shows the feature evolution, whereas the first plot in the middle layer represents the toy dataset. The bottom two plots show the mapping to the input space. The left side is our method, and the right side is NODE.

Generally, there is a class of functions that a NODE cannot model (Zhang et al., 2020). Let $\mathcal{X} = \mathbb{R}^p$ and $\mathcal{Z} \subset \mathcal{X}$ be a set that partitions $\mathcal{X}$ into two or more disjointed, connected subsets $\mathcal{C}_i$, for $i = [m]$; that is, $\mathcal{X} = \mathcal{Z} \cup (\bigcup_i \mathcal{C}_i)$. Then, no NODEs can model a mapping $h : \mathcal{X} \to \mathcal{X}$ that satisfy both (1) $h$ is an identity transformation on $\mathcal{Z}$, i.e., $\forall z \in \mathcal{Z}, h(z) = z$ and (2) $h$ maps some $x \in \mathcal{C}_i$, for $i \neq j$.

Reasoning that the trajectories cannot cross at the same moment, we intend to add the factor of discretization before the continuous trajectory. This discretization makes the trajectory no longer subject to the effect of not being able to cross over. If the trajectory does cross at some point, then the crossover will be achieved by the oscillator. Based on this idea, we introduce the Oscillation NODEs (ONODEs). It solves the problem that NODEs' trajectories cannot be crossed, thus it is a universal approximator and we provide proof in Theorem 4.1. In addition, it has fewer NFEs and faster convergence speed, which we will discuss in Section 4.2.

Specifically, an oscillator $O : \mathbb{R}^d \to \mathbb{R}^d$ approximates function $g : \mathcal{X} \to \mathcal{X}'$. The oscillator $O$ is parameterized by a perceptron with a single hidden layer and acts on the input data $\mathbf{x}_0$. The output $g(\mathbf{x}_0)$ and input $\mathbf{x}_0$ of the oscillator will have the same dimension $\mathbb{R}^d$. A formal description is as follows:

$$h(t) = g(\mathbf{x}_t, \theta_g), \quad h'(t) = f(h \circ g(\mathbf{x}_t, \theta_g), t, \theta_f), \quad h(0) = g(\mathbf{x}_0, \theta_g), \tag{5}$$

where the $\mathbf{x}_0$ is the input data, $h(t)$ is the function we want to estimate. The function $f$ is the first order derivative equation of $h$ with respect to time $t$, and $\theta_f$ represents the parameters in $f$. The $\theta_g$ represents the parameters in $g$. Figure 2 shows visually how our model works through a toy example. We show the proof of our method (Equation 5) is a universal approximator in Theorem 4.1, which is motivated by Kidger (2022). Before the proof, we give two definitions to represent the space of continuous functions and nonlinear functions, respectively.

**Definition 4.1** *Let $cont(X \to Y)$ be the space of continuous functions $X \to Y$ (with respect to some topologies on $X$ and $Y$).*

**Definition 4.2** *Let $nonl(X \to Y)$ be the space of nonlinear functions $X \to Y$.*

**Theorem 4.1** *(ONODEs are universal approximators) Let $d \in \mathbb{N}$. For $g \in nonl(\mathbb{R}^d \to \mathbb{R}^d)$, and $f \in cont(\mathbb{R} \times \mathbb{R}^d \to \mathbb{R}^d)$ which has a unique solution, let $\phi_{g,f} : \mathbb{R}^d \to \mathbb{R}^d$ denote the map $\mathcal{X} \to \mathcal{X}'$ with $h(t), h'(t)$ and $h(0) = g(\boldsymbol{x}_0, \theta_g)$, for $t \in [0, 1]$. Take $C(\mathbb{R}^d)$ to be the family of real functions that one wishes to approximate in $\mathbb{R}^d$ and $span\{\phi_{g,f}\}$, then $span\{\phi_{g,f}\}$ is dense in $C(\mathbb{R}^d)$.*

The proof of theorem 4.1 is shown in APPENDIX A.1.

## 4.2 ONODEs have a higher computational efficiency

Besides our proposed ONODEs, there exists another augmented method that makes a NODE a universal approximator, i.e., ANODEs (Dupont et al., 2019). The ANODEs add extra dimensions to the NODEs for the input, thus changing the input from $h(0) \in \mathbb{R}^d$ to $\begin{bmatrix} h(0) \\ \mathbf{0} \end{bmatrix} \in \mathbb{R}^{d+a}$. It bypasses the trajectory crossing problem by adding extra dimensions of $\mathbb{R}^a$. However, this does not essentially change the problem that node trajectories cannot cross. Even in a high-dimensional space, trajectories still cannot cross at the same time. The high-dimensional space, however, causes a rise in degrees of freedom from $d$ to $d + a$, leading to an increase in the parameter space and theoretical time complexity of estimating the vector field.

Our approach, on the other hand, allows the trajectory to "jump" between adjacent time points by adding the oscillator $g$. The degree of its oscillation is controlled by the parameters $\theta_g$ in $g$. Thus, when given an input and an output, we do not have to optimize the vector field in the original space $\mathcal{X} \in \mathbb{R}^d$, because the vector field $\mathcal{V}$ in $\mathcal{X}$ can be extremely complex. We can optimize the vector field in another space of the same dimension, i.e., $\mathcal{X}' \in \mathbb{R}^d$, in which the vector field $\mathcal{V}'$ may be simple and easy to be optimized. The trajectories based on this vector field $\mathcal{V}'$ will also become simple.

To further illustrate the time efficiency of our method, we consider its training process. For a NODE, its forward evaluation and backpropagation can be calculated by using a black-box differential equation solver, called ODE solvers (Kutta, 1901; Hairer, 1987; Runge, 1895). The *Adjoint Sensitivity Method* (Pontryagin et al., 1961) is used for the calculation of backpropagation. It computes gradients by solving a second, augmented ODE backward in time, and it can be applied to all ODE solvers (Chen et al., 2018).

For any ODE solvers, both fixed step size and adaptive step size solvers are often reasonable choices for NODEs. Given a final time $T$, a fixed step size solver will choose the time $t_i$ from $[0, T]$ where $\Delta t = t_{i+1} - t_i$. $\Delta t$ is fixed in advance and independent of $i$. An adaptive step solver, such as *Runge-Kutta Method* (Runge, 1895) is a relatively modern solver, which can vary the size of the next step so that the error made during the solver is approximately equal to some tolerance.

Typically in practice, a commonly used ODE solver is an adaptive step solver. Although it is not possible to estimate the absolute time theoretically for an adaptive-step-solver-based NODE, we can estimate the computational efficiency by the time complexity as well as the NFEs. Thus, we present proposition 4.1:

**Proposition 4.1** *With the same vector field modeling structure, ONODE has a higher computational efficiency than ANODE.*

Before the proof of the *proposition* 4.1, first we present *lemma* 4.1 and its proof.

**lemma 4.1** *With the same vector field modeling structure, ONODE has less time complexity as long as $N'_h < \frac{A \times N_h}{N_i} + \frac{N_i + A}{2}$, where $N'_h$ is the number of neurons in the hidden layer of $g$, $N_h$ is the number of neurons in the hidden layer of $f$, $A$ is the augmented dimensions, and $N_i$ is the input dimensions.*

The proof of lemma 4.1 is shown in APPENDIX A.2.

Since the adaptive step solvers such as *Runge-Kutta* are used in both ANODEs and ONODEs, we cannot obtain *Proposition* 4.1 immediately from *lemma* 4.1. However, since the NFEs are equal to the number of iterations of $f$ in one training iteration and we experimentally show that an ONODE requires much fewer NFEs than an ANODE, thus we can obtain *Proposition* 4.1.

It is worth noting that our method can still be augmented by adding additional dimensions. But the nature of our method differs from ANODE in that the trajectories of our method are crossable in the original input space $\mathcal{X} \in \mathbb{R}^d$. However, in the extra space $\mathbb{R}^a$, its trajectories remain uncrossable, which is inherited from an ANODE. Formally, an ONODE with the extra dimensions will be:

$$h(t) = \begin{bmatrix} g(\mathbf{x}_t, \theta_g) \\ \mathbf{a}(t) \end{bmatrix}, \quad h'(t) = f(\begin{bmatrix} h(\mathbf{x}_t, \theta_g) \\ \mathbf{a}(t) \end{bmatrix}, t, \theta_f), \quad h(0) = \begin{bmatrix} g(\mathbf{x}_0, \theta_g) \\ \mathbf{0} \end{bmatrix}, \tag{6}$$

where $\mathbf{a}(t)$ represents the extra dimensions for the original input. The function $g$ represents the oscillator with parameters $\theta_g$. The augmented dimension can be seen as an extra hyperparameter to tune. When the extra dimension is set to $0$, it becomes the ONODEs in Equation 5. For a more intuitive view, we show how our method works with a toy example and compares it with the NODE and the ANODE in Figure 3.

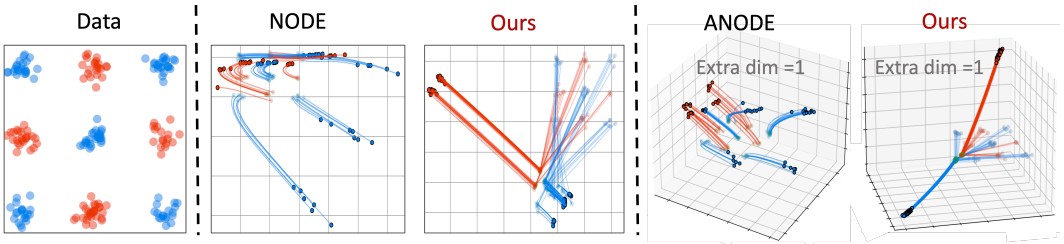

Figure 3: From left to right: A toy dataset, the NODE, our method, the ANODE with one extra dimension, our method with one extra dimension.

We show that the ONODE with extra dimensions is also a universal approximator in *Theorem* 4.2.

**Theorem 4.2** *(ONODEs with extra dimensions are universal approximators) Let $d, a \in \mathbb{N}$. For $g \in nonl(\mathbb{R}^d \to \mathbb{R}^d)$, $l \in nonl(\mathbb{R}^d \to \mathbb{R}^{d+a})$, and $f \in cout(\mathbb{R} \times \mathbb{R}^{d+a} \to \mathbb{R}^{d+a})$ which has a unique solution, let $\phi_{g,l,f} : \mathbb{R}^d \to \mathbb{R}^{d+a}$ denote the map $\mathcal{X} \to \mathcal{X}'$ with $h(t) = \begin{bmatrix} g(\boldsymbol{x}_t, \theta_g) \\ \boldsymbol{a}(t) \end{bmatrix}$, $h'(t) = f(\begin{bmatrix} h(\boldsymbol{x}_t, \theta_g) \\ \boldsymbol{a}(t) \end{bmatrix}, t, \theta_f)$, $h(0) = \begin{bmatrix} g(\boldsymbol{x}_0, \theta_g) \\ \boldsymbol{0} \end{bmatrix}$, for $t \in [0, 1]$. Take $C(\mathbb{R}^{d+a})$ to be the family of real functions that one wishes to approximate in $\mathbb{R}^{d+a}$ and set $\sum_d = span\{\phi_{g,l,f}\}$, then $\sum_d$ is dense in $C(\mathbb{R}^{d+a})$.*

The proof of theorem 4.2 is shown in APPENDIX **??**.

It is important to note that the extra dimensions change the input space which, depending on the application, may not be desirable. We experimentally demonstrate that with or without adding extra dimensions, our model is always better than the original NODEs and ANODEs. In many cases, our model without extra dimensions is better than the ANODEs which have extra dimensions.

## 5 EXPERIMENTS

We will demonstrate the superiority of our model in terms of accuracy, the number of function evaluations (NFEs), and convergence speed. In Section 5.1, we introduce the datasets and environment settings. In Section 5.2 we show on two toy datasets and three real-life image datasets that our model has better prediction accuracy, fewer NFEs, and faster convergence speed compared to baselines. In Section 5.3 we apply our model to the extrapolating time series tasks. We also illustrate that the model has less prediction error and a higher computational efficiency compared to baselines.

### 5.1 ENVIRONMENT SETUP

**Datasets.** We evaluated our model with two toy datasets and three image datasets in the classification task. We evaluated our model using five mathematical functions in the extrapolating time series

task. Demonstrations of the toy datasets are shown in Figure 2 and Figure 3, respectively. The first toy dataset consists of four concentric circles, each consisting of $1,000$ randomly generated data points. The second toy dataset consists of nine equally spaced data stacks, each consisting of $1,000$ randomly generated points based on a Gaussian distribution. For the image classification task, we evaluate our model on the MNIST (Deng, 2012), CIFAR-10, and CIFAR-100 datasets (Krizhevsky et al., 2009). MNIST is a handwritten digit database with a training set of $60,000$ examples. The CIFAR-10 training dataset consists of $60,000$ $32 \times 32$ color images in ten classes, and CIFAR-100 has 100 classes containing 600 images each.

**Evaluation Metrics and Baselines.** For the classification task, we compared our model with NODEs and ANODEs in terms of accuracy, NFEs, and convergence speed. For the extrapolating time series problem, our baselines include SONODE (Norcliffe et al., 2020) in addition to NODE and the ANODE. We compare the training and testing mean squared error (MSE), the absolute training time, and the number of parameters. We use the number in parentheses to denote the extra dimension, e.g., ANODE (1) denotes ANODE with the extra dimension of one.

## 5.2 RESULTS ON CLASSIFICATION

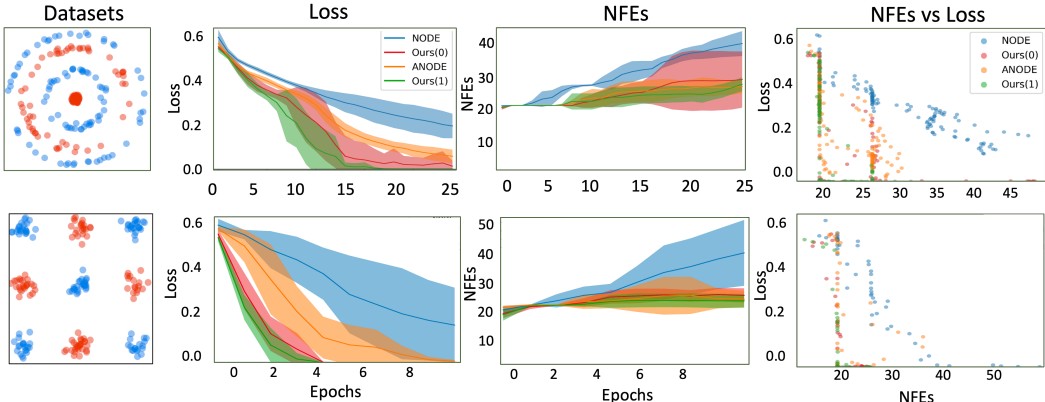

Figure 4: From left to right are: two toy datasets, loss, NFEs, and NFEs vs loss over five runs.

**Our model performs better than NODEs and ANODEs.** We first evaluated our model on two toy datasets. We model the vector field with a multilayer perceptron containing three hidden layers, each containing 16 neurons, and the activation function is *ReLU*. For the oscillator, we model it with a multilayer perceptron containing one hidden layer with 16 neurons. For our models, we use ONODE with and without one extra dimension, respectively. Our models show completely different flow trajectories and feature evolution with NODEs and ANODEs, see Figure 2 and Figure 3. Experiments show that our models have better accuracy and converge faster than NODEs and ANODEs, regardless of whether they have extra dimensions. For NFEs, our models are also much lower than NODEs and slightly lower than ANODEs. The results are shown in Figure 4. The rightmost plot indicates the relationship between NFEs and loss.

Then, we evaluated our model on the image classification task. We refer to the NODEs' modeling method for vector fields, using three convolutional layers, with the number of filters set at 128. The activation function is *ReLU*. Similarly, our oscillator modeling is also implemented by one convolutional layer. We tested our model on three image datasets, each running for 20 epochs. We recorded the validation error and average NFEs, and the results are shown in Table 1. The results show that our model is the best in both validation error and average NFEs.

**Our model has higher computational efficiency.** Since our methods only need to learn simpler vector fields and flow trajectories, they require fewer iterations to converge than NODEs and AN-ODEs. To test this, we measure the NFEs in both the forward evaluation and backpropagation process, then we obtain the total NFEs for each model. Take the CIFAR-10 for an example. As seen in Figure 5, the NFEs required by our models increase slower during both the forward evaluation and backpropagation process while it increases faster for ANODEs and NODEs. This phenomenon

Table 1: Validation errors for 20 epochs over three runs on various image datasets

|  | Validation Error | | | Average NFEs | | |
| --- | --- | --- | --- | --- | --- | --- |
|  | NODE | ANODE | Ours | NODE | ANODE | Ours |
| MNIST | $0.049 \pm 0.007$ | $0.020 \pm 0.001$ | $\mathbf{0.019 \pm 0.002}$ | $240.01 \pm 43.46$ | $166.97 \pm 110.33$ | $\mathbf{72.74 \pm 0.71}$ |
| CIFAR-10 | $0.456 \pm 0.002$ | $0.435 \pm 0.008$ | $\mathbf{0.410 \pm 0.003}$ | $65.48 \pm 1.92$ | $62.46 \pm 1.98$ | $\mathbf{59.80 \pm 1.93}$ |
| CIFAR-100 | $0.641 \pm 0.003$ | $0.620 \pm 0.002$ | $\mathbf{0.618 \pm 0.005}$ | $82.96 \pm 2.13$ | $75.37 \pm 0.59$ | $\mathbf{67.54 \pm 3.34}$ |

is particularly evident for the backpropagation process, where our method requires roughly $\frac{2}{3}$ of the NFEs of ANODEs and roughly $\frac{1}{2}$ of the NODEs. We obtain similar results for other image datasets such as MNIST and CIFAR-100.

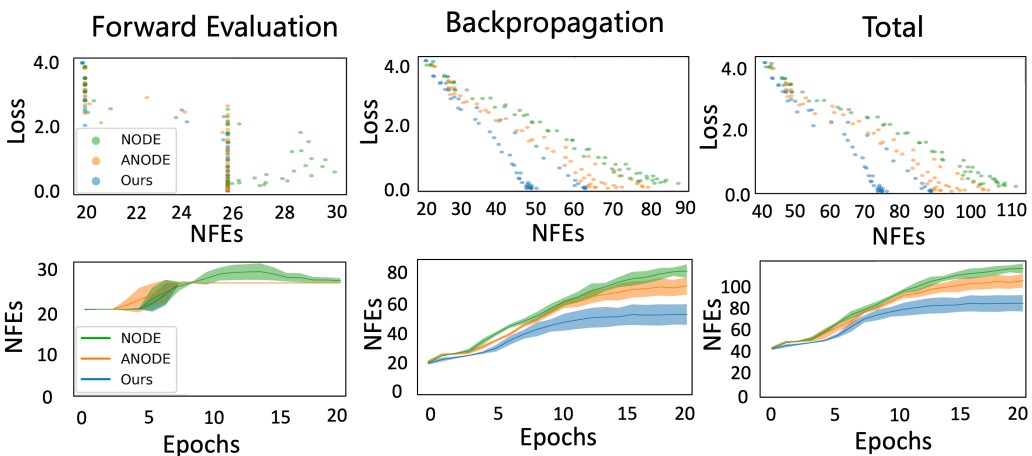

Figure 5: Based on the CIFAR-10, the three plots in the first row show the relationship between NFEs and loss in the forward evaluation, backpropagation, and total training process, respectively. The three plots in the second row show the trend of NFEs as epochs increase.

## 5.3 RESULTS ON EXTRAPOLATING TIME SERIES

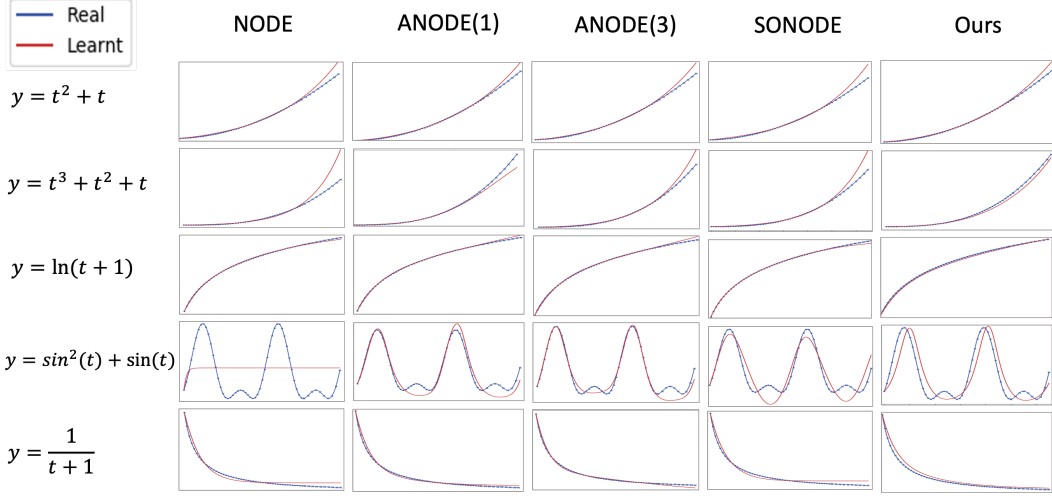

Figure 6: Fitting and extrapolation results of different models for five time series functions.

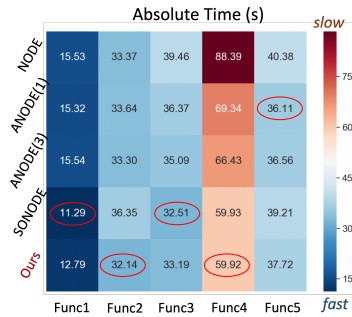

Figure 7: The training absolute time for each model in 500 iterations under the exact same setting.

We investigate the ability of our model to fit and extrapolate time series. We used five functions, as shown in Figure 6. Each function has 50 uniformly sampled points in 0 to 10 seconds as the training set, followed by 10 points in 3 seconds as the predicted values. Our model is set to have one extra dimension. The baselines were chosen as NODE, ANODE with one extra dimension, ANODE with three extra dimensions, and SON-ODE. Each model was trained for 500 iterations in the same environment. First, we compared the absolute training time (Figure 7) for each model. We found that our model has a faster training speed with SONODE. Then, we compared the training MSE, test MSE, and the number of parameters, as shown in Figure 8. For visual comparison, we have ranked the performance of the models according to MSE from smallest to largest. Our model has the smallest value in most of the training MSE and all the test MSE. We also found that as the extra dimension increases, there is no guarantee that ANODE becomes better. Our model has about the same number of parameters as ANODE, but has much smaller than the number of parameters of SONODE.

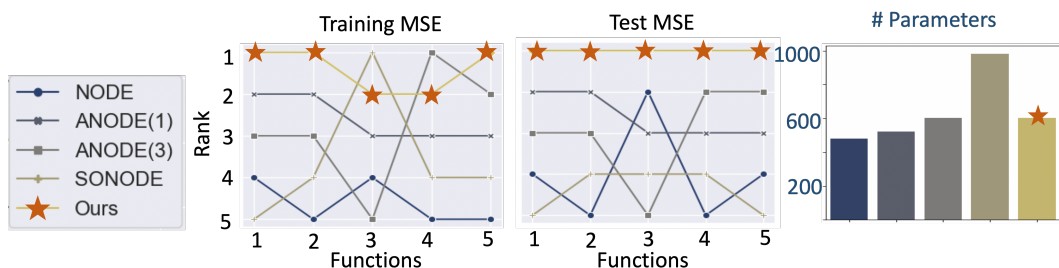

Figure 8: The middle two plots show the five models ranked in order of MSE from smallest to largest. The rightmost plot shows the total number of parameters for each model.

## 6 CONCLUSION

We propose the ONODE, which is an oscillator-based continuous deep learning model. Due to the presence of an oscillator, the trajectories of our model can "jump", making it to be a universal approximator. We also demonstrate that the ONODE is computationally efficient. We validate our model through various experiments including classification tasks and extrapolation on time series. Compared to the baselines, our model achieves the best results on a variety of metrics.

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

## A APPENDIX

### A.1 THE PROOF OF THEOREM 4.1

*Proof*: Suppose the input $x \in \mathbb{R}^d$, the system of ODEs will solve a series of equations $y_0, y_1, ..., y_M$ as follows: (1) $h_0(0) = g(x) \in \mathbb{R}^d$ and $\frac{dh_0}{dt}(t) = 0$, (2) $h_1(0) = 0 \in \mathbb{R}^{d \times d}$ and $\frac{dh_1}{dt} = h_0(t) \otimes h_0(t)$, (3) $h_2(0) = 0 \in \mathbb{R}^{d^3}$ and $\frac{dh_2}{dt}(t) = h_1(t) \otimes h_0(t)$, (4) $h_3(0) = 0 \in \mathbb{R}^{d^4}$ and $\frac{dh_3}{dt}(t) = h_2(t) \otimes h_0(t)$, ... , (M+1) $h_M(0) = 0 \in \mathbb{R}^{d^{M+1}}$ and $\frac{dh_M}{dt}(t) = h_{M-1}(t) \otimes h_0(t)$. The solution will be written immediately: (1) $h_0(t) = g(x)$, (2) $h_1(t) = \int_0^t g(x) \otimes g(x) ds = tg(x)^{\otimes 2}$, (3) $h_2(t) = \int_0^t g(x)^{\otimes 2} t \otimes g(x) ds = \frac{1}{2} t^2 g(x)^{\otimes 3}$, (4) $h_3(t) = \int_0^t g(x)^{\otimes 3} t \otimes g(x) ds = \frac{1}{6} t^3 g(x)^{\otimes 4}$, ... , (M+1) $h_M(t) = \int_0^t g(x)^{\otimes (M+1)} t \otimes g(x) ds = \frac{1}{M!} t^M g(x)^{\otimes (M+1)}$. Set $t = 1$ and we will obtain the collection in $g(x) \in \mathbb{R}^d$ up to degree $M + 1$, i.e., $\{g(x), g(x)^{\otimes 2}, \frac{1}{2} g(x)^{\otimes 3}, \frac{1}{6} g(x)^{\otimes 4}, ..., \frac{1}{M!} g(x)^{\otimes (M+1)}\}$. The function $g(x)$ is realized by a single hidden layer perception utilizing a Relu active function. Assume $\mu$ is a non-negative finite measure on $\mathbb{R}^d$ with compact support, and continuous with respect to the Lebesgue measure. Then $\sum_d$ is dense in $L^p(\mu), 1 \le p < \infty$, if and only if, the non-linearity in $g$ is not a polynomial, where $L^p(\mu)$ is the set of all measurable functions f such that $\| f \|_{L^p(\mu)} = (\int_{R^d} |f(x)|^p d\mu(x))^{1/p} < \infty$ (Leshno et al., 1993; Hornik et al., 1989; Hornik, 1991; Pinkus, 1999). Thus, $span\{\phi_{g,f}\}$ is dense in $C(\mathbb{R}^d)$ and it is a universal approximator.

### A.2 THE PROOF OF LEMMA 4.1

*Proof:* For a fair comparison between ONODE and ANODE, we assume that the input and output dimensions are the same, i.e., $N_i = N_o$, where the $N_o$ represents the output dimensions. For the vector field modeling structure, we assume there exist two hidden layers with each hidden layer having $N_h$ neurons. For convenience, we presume that there is only one sample. For forward evaluation, an ANODE will have the time complexity of $\mathcal{O}^f_{Anode} = \mathcal{O}((N_i + 1 + A)N_h) + \mathcal{O}(N_h N_h) + \mathcal{O}(N_h(N_i + A)) + \mathcal{O}((N_i + A)N_i) = \mathcal{O}(2N_i N_h + 2AN_h + N_h^2 + N_h + N_i^2 + AN_i)$. Our proposed ONODE will have the time complexity of $\mathcal{O}^f_{Onode} = \mathcal{O}(N_i N_h') + \mathcal{O}(N_h' N_i) + \mathcal{O}((N_i + 1)N_h) + \mathcal{O}(N_h N_h) + \mathcal{O}(N_h H_i) = \mathcal{O}(2N_i N_h + N_h^2 + N_h + 2N_i N_h')$. Since the backpropagation has the same complexity as the forward evaluation, the total time complexities in one iteration for ANODE and ONODE are $\mathcal{O}_{Anode} = 2\mathcal{O}^f_{Anode}$ and $\mathcal{O}_{Onode} = 2\mathcal{O}^f_{Onode}$, respectively. We can immediately get the solution that $\mathcal{O}_{Onode} < \mathcal{O}_{Anode}$ as long as $N_h' < \frac{A \times N_h}{N_i} + \frac{N_i + A}{2}$.

### A.3 THE PROOF OF LEMMA 4.2

*Proof:* The proof is similar to that of proof A.1. Since the extra dimensions $\mathbb{R}^a$ will be added into the ONODE through a nonlinear function $l$, it will not change the result in proof A.1. Thus, we obtain the result that $span\{\phi_{g,l,f}\}$ is dense in $C(\mathbb{R}^{d+a})$.

