# OpenReview forum: "Oscillation Neural Ordinary Differential Equations"
_ICLR.cc/2023/Conference — Submitted to ICLR 2023_

### Official Review · Reviewer_a9dv · 2022-10-22

**Confidence:** 3
**Correctness:** 4
**Technical Novelty And Significance:** 2
**Empirical Novelty And Significance:** 2
**Recommendation:** 3

**Clarity, Quality, Novelty And Reproducibility:**

The overall quality of the paper is good. The writing is clear, and it is easy to follow the arguments that are made by the authors. However, I would suggest moving the proofs into the appendix, since they are very dense and painful to read in their current form. This would leave room for a better motivation of the proposed innovation and for more experiments and comparisons with other methods. Due to the simplicity of the proposed modification, it is easy to reproduce the results.

**Strength And Weaknesses:**

This paper proposes a very simple modification that leads to a significant effect, i.e., the resulting model is a universal approximator. However, this result is not very surprising since the term that is introduced can be seen as some form of a gating function if a sigmoid is wrapped around the function $g$. Further, the results presented are interesting but limited. Here are more detailed comments that I would like to see addressed:

The way you first introduce $g$ is a bit mysterious. First, it would be nice to have an extended discussion on oscillation ODEs (I am not very familiar with these types of systems, and I assume most readers are neither). Where do such ODEs appear in the real-world (I assume there are many interesting examples), and what are their drawbacks and limitations? I assume that ODEs that have highly oscillatory solutions have stability issues and/or cannot be solved efficiently using standard numerical integrators. So why are these models good for motivating a new neural network architecture (is that because jumps are introduced)? Next, a good discussion of how to design / model $g$ is missing. Later in the text, after you prove your theorem, you briefly state that `the function g(x) is realized by a single hidden layer perception utilizing a Relu active function.’ Why do you use Relu here? It seems to be a bit of an unnatural choice, because you don’t want that the activations of $g$ can grow too much, or am I missing something? I would choose a tanh or sigmoid function around $g$ so the activations are bounded or does $g$ need to be some unbounded function in order that your theorem holds (I don’t think so)?

The experiments are weak for two reasons:
* Only ANODE is used for comparison in 5.2. I miss a more detailed discussion and the comparison with coupled NODEs / second-order ODEs. Will the proposed model perform better or like these more general formulations on the tasks considered? Moreover, a discussion and comparison to Neural Dealy Differential Equations [1] is missing.

* The considered set of experiments is limited. It is good to use some toy tasks to demonstrate the advantage and to build some intuition. But I would also like to see how the model performs on more interesting (challenging) real-world tasks.

* Moreover, the tables need to list the parameter counts of the different models.

* Which ranges for the tuning parameters did you explore, and did you use a validation set for tuning in addition to the final test set?


[1] Neural Delay Differential Equations. ICLR 2021.

**Summary Of The Paper:**

This paper is proposing a modified neural ordinary differential equation (NODE) that augments the function that describes the vector field with an additional term that introduces oscillations into the dynamics. The authors show that this modification can overcome limitations of standard NODEs by (i) proofing that the new model is a universal approximator, and (ii) by the means of experiments on classification and time series extrapolation tasks.

**Summary Of The Review:**

In summary, the paper presents some neat results, but I don’t feel that these results are significant enough to accept this paper. That is, because the modification is incremental and the experiments do not show that the proposed model leads to any substantial improvement compared to ANODE (e.g., improving the test error for CIFAR-100 from 0.62 to 0.618 is uninteresting). In the current form the paper is a good workshop paper.

---

> ### Author Response · Authors · 2022-11-19
> **Rebuttal**
>
> Thanks for your comments. We carefully consider your suggestion. We submit a modified version, please see the Rebuttal revision. We will do more in future work.
>
> The g is a function to make the NODE a universal approximator. The way g does this is by jumping.
> We choose Relu because the Relu function works better in our experiments and can avoid gradient disappearance as much as possible.
> For the part of the experiment:
> We compared our model with NODEs, ANODEs, and coupled NODEs in  5.3 extrapolating time series. Since the typical use of coupled NODEs is time series, we did not compare it in the classification part in 5.2.

---

### Official Review · Reviewer_p8TZ · 2022-10-23

**Confidence:** 4
**Correctness:** 2
**Technical Novelty And Significance:** 3
**Empirical Novelty And Significance:** 2
**Recommendation:** 3

**Clarity, Quality, Novelty And Reproducibility:**

There were some parts of the paper that I found difficult to understand. It is necessary to clarify some mathematical notations that are confusing or misleading.

**Strength And Weaknesses:**

**Strength**

The main idea of the paper is interesting and important.

**Weaknesses**

1) The function  $g: \mathcal{X} \longrightarrow  \mathcal{X}^{\prime}$ defined in eq.(5) is not clear enough. $\mathcal{X}$  and $ \mathcal{X}^{\prime}$ need to be defined as well.

2) The authors have mentioned that their approach allows the trajectory to “jump” between adjacent time points by adding the oscillator $g$. However, it is unclear how this can be achieved neither through their explanation nor eq. (5).

3) In Theorem 4.1/ Theorem 4.2, it is mentioned: "$f \in cont(\mathbb{R} \times \mathbb{R}^d → \mathbb{R}^d)$/$f \in cont(\mathbb{R} \times \mathbb{R}^{d+a} → \mathbb{R}^{d+a})$ which has a unique solution". What do you mean by the *unique solution* of $f$ in both theorems?

4) By $C(\mathbb{R}^d)$, do you mean the set of all the real-valued **continuous** maps on $\mathbb{R}^d$?

5) In the proof of Theorem 4.1, it is proven that $\sum_d$ is dense in $L^p(\mu)$. But, how does this imply that $\sum_d$ is also dense in $C(\mathbb{R}^d)$ (especially since $\mathbb{R}^d$ is not compact)?

6) In the proof of Theorem 4.1, it is mentiond: "Assume $\mu$ is a non-negative finite measure on $\mathbb{R}^d$ with compact support, and continuous with respect to the Lebesgue measure".  What are sufficient and necessary conditions for having such a measure?

7) On page 5, section 4.2, it is written: " [...] the vector field in the original space $X \in \mathbb{R}^d$, [...]". What do you mean by the
**space**  $X \in \mathbb{R}^d$? Likewise, what does $X^{\prime} \in \mathbb{R}^d$ mean?

8) On page 4, section 4.1, by $\phi_T$ do you mean the flow of an ODE?


**Summary Of The Paper:**

A universal approximator is proposed in this paper using an oscillator NODE (ONODE) whose trajectories can "jump". It was shown that the ONODE is computationally efficient. A number of experiments (including classification tasks and extrapolation on time series) were conducted to validate the model.

**Summary Of The Review:**

Overall, I think this paper need more theoretical support. Some claims are not well-supported and need to be proven.

---

> ### Author Response · Authors · 2022-11-19
> **Rebuttal**
>
> Thanks for your comments. We modify the math part of our paper and try to make it clearer, please see the rebuttal version.

---

> > ### Comment · Reviewer_p8TZ · 2022-12-07
> > **Thanks!**
> >
> > Thank you for your response! However, my concerns have not yet been addressed.

---

### Official Review · Reviewer_nasm · 2022-10-23

**Confidence:** 3
**Correctness:** 2
**Technical Novelty And Significance:** 2
**Empirical Novelty And Significance:** 1
**Recommendation:** 3

**Clarity, Quality, Novelty And Reproducibility:**

Clarity: The paper is mostly clear but the figures need to be revised.

Quality: The discussion of prior work in the field is lacking, and the experiments are rather narrow and not sufficient to draw conclusions.

Novelty: The method seems to be new.

Reproducibility: From the text of the paper I could not reproduce the results. Crucial details like the chosen architecture and hyper parameters are not provided in the text.



**Strength And Weaknesses:**

Strengths:

1. The paper identifies three main challenges in current ODEs and proposes a method to alleviate them.

2. Empirically the method seems to work.

Weaknesses:

1. The figures quality and clarity need to be improved. They do not look professional and are hard to understand.

2. One of the problems that the authors state that exists in current Neural ODEs is "The reason
for the second limitation is caused by the straightforward optimization approach of the vector field.
There is no guarantee that learning the vector field is a better choice than learning the estimated functions
directly. Sometimes it will need many function evaluations to optimize the vector field, so the
difficulties go beyond learning the estimated functions themselves" in page 1. However no citations or self explanation is given in the paper.

3. The authors are lacking reference and discussion of prior work, for example "Stable Architectures for Deep Neural Networks".

4.  I am not sure why the authors call the method 'oscillatory'. Typically this is a term that is reserved to describe hyperbolic or wave like dynamical systems. However here the authors propose to introduce 'jumps' into the trajectories.

5. The experimental scope is rather narrow and it is hard to draw conclusions from experiments on small datasets like CIFAR-10 and CIFAR-100. I believe that experiments with larger and standard datasets like ImageNet can be more convincing. Also, the authors should compare their obtained accuracy with other CNNs that are not necessarily ODE based, for a fair comparison.

6. A question to the authors: Why do you report the *validation* accuracy on CIFAR-10 and CIFAR-100 ? the norm is to report the test accuracy.




**Summary Of The Paper:**

The paper suggests adding jumping terms to the training of Neural ODEs to improve training and computational time.
The conduct an analysis of the proposed model and several experimental studies.

**Summary Of The Review:**

The paper identifies problems in current Neural ODEs but fails to discuss relevant prior work and to fully evaluate their method.

---

> ### Author Response · Authors · 2022-11-19
> **Rebuttal**
>
> Thanks for your comments.
> For weakness 1, we improve the figures in our paper.
> For weakness 2, it is a conjecture and we prove it through experiments. We modify the statement in the paper.
> For weakness 4, the reason we call our method "oscillatory" is that trajectories can cross in jumps (or oscillations).
> For weaknesses 3,5 and 6, we will improve in future work.

---

### Decision · Program_Chairs · 2023-01-20

**Decision:**

Reject

**Justification For Why Not Higher Score:**

N/A

**Justification For Why Not Lower Score:**

N/A

**Metareview: Summary, Strengths And Weaknesses:**

All reviewers are in agreement that the work is not ready for publication. In particular, the reviewers' concerns were not addressed during the discussion period.

I advise the authors to take the detailed feedback provided by the reviewers and revise the paper accordingly. I vote for rejection.

**Summary Of Ac-Reviewer Meeting:**

N/A